# Dietary Supplementation with Protocatechuic Acid and a Complex *Eucommia ulmoides* Leaf Extract Differentially Enhanced Reproductive Performance and Modulated Gut Microbiota in Late-Pregnancy Sows

**DOI:** 10.3390/ani15213166

**Published:** 2025-10-31

**Authors:** Jijun Tan, Jianhua He, Hongfu Zhang, Shusong Wu

**Affiliations:** 1Department of Animal Nutrition and Feed Science, School of Life Sciences and Environmental Resources, Yichun University, Yichun 336000, China; jijun995@jxycu.edu.cn; 2Hunan Collaborative Innovation Center for Utilization of Botanical Functional Ingredients, Hunan Agricultural University, Changsha 410128, China; 3State Key Laboratory of Animal Nutrition, Institute of Animal Sciences, Chinese Academy of Agricultural Sciences, Beijing 100193, China

**Keywords:** protocatechuic acid, *Eucommia ulmoides*, sow, late pregnancy, reproductive performance

## Abstract

**Simple Summary:**

Fatness management in late-pregnancy sows is critical for optimizing reproductive performance and subsequent productivity, as it is closely associated with maternal metabolic status. Given the essential roles of gut microbiota in host metabolic pathways, microbiota-related research has received increasing attention over the past few decades. Our previous study demonstrated that protocatechuic acid (PCA) significantly improved glycolipid metabolism in a murine model, but relevant applications of PCA in sows remain largely unexplored. To provide a scientific basis for the rational utilization of PCA and PCA-rich natural resources, this study aims to investigate the effects of dietary supplementation with PCA and PCA-rich *Eucommia ulmoides* leaf extract (EU) on reproductive performance in late-pregnancy sows. Results in this paper showed that dietary supplementation with PCA and PCA-rich EU enhanced reproductive performance and colostrum immunoglobulin levels in late-pregnancy sows, which were potentially mediated through alteration of maternal gut microbiota linked to inflammation and glucose metabolism. This original manuscript provides scientific references for the rational utilization of PCA, EU, and other PCA-rich natural resources in sow feeding programs.

**Abstract:**

Nutrition during late pregnancy plays a critical role in fetal development. This study was conducted to investigate the effects and underlying mechanisms of protocatechuic acid (PCA) and PCA-rich *Eucommia ulmoides* leaf extract (EU) on reproductive performance using late-pregnancy sows as a model. A total of 30 sows (Landrace × Yorkshire; average parity: 3–4) with similar body condition (assessed as a score of 4 on a 5-point scale) were randomly assigned to three treatments (*n* = 10 per group) from day 80 of gestation until farrowing and fed either a basal diet, a basal diet supplemented with 200 g/t of PCA, or 1000 g/t of EU. Results demonstrated that dietary supplementation with PCA and EU, which delivered a negligible amount of PCA but contained other bioactive phytochemicals such as chlorogenic acid, significantly increased litter weight at birth and the number of healthy piglets (*p* < 0.05), along with elevated levels of colostral immunoglobulins and reduced serum interleukin (IL)-6 concentrations (*p* < 0.05). Furthermore, PCA supplementation was associated with a decrease in fasting glucose levels and improved insulin sensitivity (*p* < 0.05), accompanied by an increased relative abundance of Bacteroidetes (*p* < 0.05). Analysis of gut microbial composition revealed that both PCA and EU reduced the relative abundance of *Paraprevotella* (*p* < 0.05), while PCA increased the abundance of the *dgA11_gut_group* (*p* < 0.05), and EU enriched *Caldicoprobacter* (*p* < 0.05). Correlation analysis indicated that PCA- and EU-modulated genera, such as *Lysinibacillus*, were positively associated with colostrum lactose and colostrum fat but negatively correlated with the number of somatic cells, colostrum protein, degreased dry matter, total solids, and urea nitrogen (*p* < 0.05). In conclusion, dietary supplementation with PCA and EU differentially enhanced reproductive performance and colostrum immunoglobulins, at least partially, through the modulation of inflammation and glucose metabolism-related gut microbiota in late-pregnancy sows.

## 1. Introduction

Late pregnancy represents a critical developmental window for the fetus, particularly in mammalian species [1]. During this period, the maternal physiological status undergoes dynamic biochemical changes that influence fetal development via placental transfer [2]. Maintaining glycolipid metabolic homeostasis during late pregnancy is essential for optimal maternal performance, as multiple tissues compete for plasma glucose to support mammary gland development, fetal growth, milk synthesis, and uterine contractions [3]. Physiological insulin resistance facilitates the redistribution of glucose to support both mammary and fetal development. However, persistent and excessive insulin resistance, referred to as pathological insulin resistance, can have detrimental effects on both mother and fetus [4]. A clinical study involving 2647 women with gestational diabetes mellitus in late pregnancy revealed that increased insulin resistance is predictive of adverse pregnancy outcomes, including cesarean delivery, preterm birth, and macrosomia [5]. Colostrum-derived immunoglobulins are crucial for the establishment of neonatal immunity [6], yet maternal hyperglycemia-induced immunostimulation in colostrum [7] has been associated with unfavorable metabolic profiles in offspring, including elevated biomarkers of insulin resistance [8]. Maternal obesity is linked to systemic inflammation and may expose the developing fetus to inflammatory environments [9]. Research has indicated that obesity in sows promotes placental lipotoxicity, oxidative stress, and inflammation, potentially through activation of the JNK/NF-κB signaling pathway [10]. As an emerging research focus [11,12,13], gut microbiota play pivotal roles in ameliorating insulin resistance and promoting placental angiogenesis [14], and can profoundly influence maternal and offspring health through vertical transmission [15]. Crusell et al. [16] observed a decline in fecal microbial OTUs (*p* = 0.0002) and Shannon’s diversity (*p* = 0.012) from late pregnancy to postpartum in a cohort of 125 pregnant women (43 women with gestational diabetes mellitus and 82 women with normal glucose), suggesting a potential link to heightened inflammatory responses and insulin resistance [17]. Alterations in gut microbiota during late pregnancy are closely associated with the absorption and metabolism of microbiota-derived metabolites such as glucose, thereby contributing to the formation of a specialized metabolic system that supports maternal tissue and fetal development [18].

Although dietary restriction is a commonly used and cost-effective strategy for managing body condition in pregnant sows [19], natural feed additives offer promising alternatives that reduce the need for high-level technical expertise in on-farm management [20]. Polyphenol-rich wild plants are widely consumed globally [21,22]. Anthocyanins, due to their catecholic structure, exhibit potent antioxidant properties and hold promise as therapeutic agents for metabolic disorders, as demonstrated in rodent models and human clinical trials [23]. Our previous studies have demonstrated that anthocyanins, particularly cyanidin-3-glucoside (C3G), derived from blue honeysuckle (*Lonicera caerulea* L. berry), can alleviate inflammation [24] and improve glucose metabolism [25] in high-fat diet (HFD)-induced obese mice. Protocatechuic acid (PCA), a phenolic metabolite of C3G found in numerous plant species, especially in the endemic Chinese herb *Eucommia ulmoides* native to Zhangjiajie, Hunan Province, has been shown to improve inflammation and insulin resistance in mice by upregulating *fibroblast growth factor 1* (*Fgf1*), *insulin-like growth factor binding protein 2* (*Igfbp2*), *insulin receptor substrate 1* (*Irs1*), and *insulin receptor substrate 2* (*Irs2*) [26] and to modulate gut microbiota in piglets (Pig Improvement Company line 337 × C48, 28 d of age, 8.87 kg ± 0.11 kg BW) [27]. Therefore, this study was designed to investigate the effects of dietary supplementation with PCA and PCA-rich *Eucommia ulmoides* leaf extract (EU) on the reproductive performance, immune status, metabolic parameters, and gut microbiota of sows during late pregnancy.

## 2. Materials and Methods

### 2.1. Diets and Reagents

The composition of pregnancy feed is shown in Appendix A. Protocatechuic acid (PCA ≥ 97%, HPLC) was purchased from Shanghai Yuanye Bio-Technology Co., Ltd. (Shanghai, China). *Eucommia ulmoides* leaf extract (EU) was provided by Hengxing Bio-Technology Co., Ltd. (Zhangjiajie, Hunan, China).

### 2.2. High-Performance Liquid Chromatography (HPLC) Analysis

The concentration of PCA in EU was detected using an HPLC system with a C18 column (250 × 4.6 mm, JASGC, Tokyo, Japan). Briefly, EU (40 mg/mL) was dissolved in a 100% methanol aqueous solution, followed by shaking and ultrasonic treatment for at least 1 h, and ultimately filtered with a Nylon filter (13 mm × 0.22 μm, Thermo Fisher Scientific, Shanghai, China). To calculate concentration of PCA, the solvent system was a mixture of A (100% CH_3_CN) and C (0.1% HCOOH in water), and the gradient was as follows: 0–5 min (5% A; 95% C); 5–10 min (7% A; 93% C); 10–20 min (13% A; 87% C); 20–30 min (16% A; 84% C); 30–40 min (25% A; 75% C); 40–41 min (30% A; 70% C); 41–47 min (90% A; 10% C); and 47–52 min (5% A; 95% C), with the flow rate of 1.2 mL/min, 10 μL sample and column temperature of 35 °C. Spectrophotometric detection was performed at 254 nm using an Agilent 1260 Infinity II HPLC system (Agilent Technologies, Santa Clara, CA, USA). The concentration of PCA in EU was identified according to retention time with the known compounds as standards (PCA ≥ 97%, HPLC).

### 2.3. Experimental Design

The experimental procedures were approved by the Hunan Agricultural University Institutional Animal Care and Use Committee (Permission No. 2020034). A total of 30 sows (Landrace × Yorkshire; average parity: 3–4) with similar body condition (assessed as a score of 4 on a 5-point scale) were randomly allocated into 3 treatments (*n* = 10) on day 80 of pregnancy. The sample size of *n* = 10 sows per group was determined based on common practice and observed effect sizes in the prior literature investigating dietary interventions in late-pregnancy sows [28,29]. All 3 groups were fed the same basal diet (control group, CTL; for composition, see Appendix A). The PCA and EU groups were prepared by top-dressing the basal diet with the respective additive using a feed mixer immediately prior to feeding. The final diets consisted of the basal diet supplemented with 200 g/t of PCA (PCA group), or 1000 g/t of EU (EU group). Sows were housed in an environmentally controlled gestation barn (30 m × 10 m) under a 12 h light/12 h dark cycle, with ambient temperature maintained at 20 ± 2 °C and relative humidity at 60–70%. Each sow was kept in an individual stall (0.6 m × 2.1 m).

On day 110 of gestation, sows were moved to individual farrowing pens (2.2 m × 1.8 m) in a separate, environmentally controlled farrowing room. The farrowing room was maintained under the same light cycle (12 h light/12 h dark) and similar humidity (60–70%). The ambient temperature for the sows was maintained at approximately 20–22 °C, with supplemental localized heat (e.g., heat lamps) provided for the newborn piglets within the creep area of the pen. Before that, the body condition scores of sows approaching farrowing were recorded via a generic sow body condition caliper. Then, fasting auricular venous blood and colostrum samples were collected during the farrowing process (within 2 h of the first piglet being born). Moreover, reproductive performance was accurately recorded.

### 2.4. Reproductive Performance

Reproductive performance, including average individual weight, total litter size, litter weight at birth, number of healthy piglets (Landrace × Yorkshire), number of weak piglets (≤0.5 kg), number of stillbirths, and stillbirth rate in each sow, was recorded and analyzed.

### 2.5. Analysis of Colostrum Composition

Seven items in regular analysis of colostrum, including somatic cell number, colostrum fat, colostrum protein, colostrum lactose, degreased dry matter, total solids, and urea nitrogen, were achieved by the Hunan Dairy Cow Production Performance Measurement Center (Changsha, China).

The contents of immunoglobulins (IgG and IgM) in colostrum were tested using commercial swine-specific ELISA kits (Shanghai Enzyme-linked Biotechnology Co., Ltd., Shanghai, China) according to the manufacturer’s instructions. The absorbance was read using a porous chemiluminescence instrument (Varioskan flash, Thermo Fisher Scientific, Waltham, MA, USA).

### 2.6. Analysis of Serum Indexes

Serum was obtained with centrifugation at 1500× *g* for 10 min after standing at room temperature for at least 30 min. Levels of glucose, superoxide dismutase (SOD), and malondialdehyde (MDA) in serum were detected using commercial assay kits (Nanjing Jiancheng Bioengineering Research Institute, Nanjing, China) following the manufacturer’s protocols. Levels of insulin, IL-1β, and IL-6 in serum were detected, according to ELISA kits (Shanghai Enzyme-linked Biotechnology Co., Ltd., Shanghai, China). All the above indices were determined with a porous chemiluminescence instrument (Varioskan flash, Thermo Fisher Scientific, Waltham, MA, USA).

Insulin resistance was reflected with HOMA-IR (homeostasis model assessment-estimated insulin resistance) according to the following formula [30]:HOMA-IR = fasting insulinemia (mIU/L) × fasting glycemia (mg/dL)/405

### 2.7. Analysis of Gut Microbiota

Fresh fecal samples were collected from each sow via rectal stimulation on the day of farrowing (within 2 h of the first piglet being born), immediately frozen in liquid nitrogen, and stored at −80 °C until DNA extraction. Fecal DNA was extracted using a commercial Stool DNA Isolation Kit (Tiangen Biotech, Beijing, China). The quality and concentration of the extracted DNA were verified using a Nanodrop spectrophotometer (Thermo Fisher Scientific, Wilmington, DE, USA). Amplification of the bacterial 16S rRNA gene V4 region was conducted with barcoded primers 515F (5′-GTGCCAGCMGCCGCGGTAA-3′) and 806R (5′-GGACTACHVGGGTWTCTAAT-3′), as detailed in Appendix A. The PCR mixture (25 µL) contained 12.5 µL of 2× Taq PCR MasterMix, 1 µL of each primer (5 µM), 3 µL of BSA (2 ng/µL), 3 µL of DNA template (30 ng), and 4.5 µL of ddH_2_O. The thermal cycling conditions were as follows: initial denaturation at 95 °C for 5 min; 25 cycles of denaturation at 95 °C for 45 s, annealing at 50 °C for 50 s, and extension at 72 °C for 45 s, followed by a final extension at 72 °C for 10 min. Amplicons were purified, quantified, pooled in equimolar ratios, and sequenced on an Illumina platform by Allwegene Technology Inc. (Beijing, China) [26].

### 2.8. Statistical Analysis

All data are presented as the mean ± standard deviation (SD). Statistical analysis was performed using SPSS Statistics (Version 21.0, IBM Corp., Armonk, NY, USA). Differences among the 3 treatment groups were assessed with one-way analysis of variance (ANOVA). When the ANOVA indicated a significant effect, post hoc comparisons were conducted using Fisher’s Least Significant Difference (LSD) test. A *p*-value of less than 0.05 was considered statistically significant. The relationship between gut microbiota and measured indicators was evaluated using Pearson’s correlation analysis.

## 3. Results

### 3.1. Calculation of Concentration of PCA in EU

Complex substances in EU were detected at 254 nm. Among these, chlorogenic acid was the most abundant component (see Appendix A). The concentration of PCA in EU was calculated to be 0.18 mg/g by establishing a standard curve based on the standard concentration and peak area (see Appendix A).

### 3.2. Effect of PCA and EU on the Body Condition Score of Sows

As depicted in Appendix A, PCA showed a non-significant tendency to reduce the body condition score of sows (*p* = 0.062, 5.06 ± 0.80, 4.37 ± 0.92, and 4.98 ± 0.67 for the CTL, PCA, and EU groups, respectively).

### 3.3. Supplement of PCA and EU Differentially Enhanced Reproductive Performances, as Reflected by Litter Weight at Birth and the Number of Healthy Piglets

As shown in Figure 1, supplementation with PCA significantly increased the litter weight at birth (*p* < 0.05, Figure 1C) and the number of healthy piglets (*p* < 0.05, Figure 1D). A similar improvement was observed in the EU group (*p* < 0.05). However, they had limited effects (*p* > 0.05) on the average individual weight (Figure 1A), total litter size (Figure 1B), the number of weak piglets (≤0.5 kg) (Figure 1E), the number of stillbirths (Figure 1F), and the stillbirth rate (Figure 1G).

### 3.4. Supplementation with PCA and EU Differentially Increased Immunoglobulins in Colostrum

Supplementation with PCA and EU had no significant effects on seven items in the routine analysis of colostrum (*p* > 0.05, see Appendix A). Interestingly, PCA supplementation significantly elevated the levels of IgG in colostrum (*p* < 0.05, Figure 1H). Differently, EU supplementation significantly increased the levels of IgM in colostrum (*p* < 0.05, Figure 1I).

### 3.5. Effects of PCA and EU on the Redox Status, Inflammatory Cytokines, and Glucose Metabolism in Serum

As shown in Figure 2A,B, the CTL, PCA group, and EU group exhibited similar redox statuses (*p* > 0.05).

Limited effects were noted on the levels of IL-1β in serum (*p* > 0.05, Figure 2C). However, PCA supplementation significantly reduced the levels of IL-6 in serum (*p* < 0.05), and a similar reduction was observed in the EU group (*p* < 0.05, Figure 2D).

PCA supplementation during late pregnancy significantly decreased the levels of fasting glucose in serum (*p* < 0.05, Figure 2E) and improved insulin resistance (*p* < 0.05, Figure 2F). In contrast, EU supplementation had limited effects on glucose metabolism (*p* > 0.05).

### 3.6. Effects of PCA and EU on Gut Microbiota

Supplementation with PCA and EU led to greater gut microbiota diversity (see Appendix A and Figure 3A). When compared to the CTL group, closer clusters were observed between the PCA and EU groups (Figure 3B,C). Although PCA showed a non-significant trend in the ratio of Firmicutes to Bacteroidetes (*p* = 0.054, Figure 3G), it increased the relative abundance of Bacteroidetes (*p* < 0.05, Figure 3F). However, in the EU group, these changes were not significant (*p* > 0.05, Figure 3D–G). Further analysis at the genus level revealed that EU increased the relative abundance of *Caldicoprobacter* (*p* < 0.05, Figure 3H), while PCA increased the relative abundance of *dgA11_gut_group* (*p* < 0.05, Figure 3J). Compared to the CTL group, both EU and PCA decreased the relative abundance of *Paraprevotella* (*p* < 0.05, Figure 3I). All genera that were significantly correlated with the indicators were screened. Correlation analysis between gut microbiota and colostrum indicators showed that several genera positively correlated with colostrum lactose or fat had a relative decrease in the PCA and EU groups, such as *Clostridium_sensu_stricto_3* and *Lysinibacillus*. These gut microbiota were negatively correlated with the number of somatic cells, colostrum protein, degreased dry matter, total solids, and urea nitrogen. Conversely, genera that negatively correlated with colostrum lactose, including *Caldicoprobacter* and *Rikenellaceae_RC9_gut_group*, had a relative increase in the PCA and EU groups, and these gut microbiota were positively correlated with colostrum protein, degreased dry matter, total solids, and urea nitrogen (Figure 4). Correlation analyses between gut microbiota, immunoglobulins, and serum indicators revealed that genera, such as *Anaerovibrio* and *Veillonella*, positively correlated with IL-6 or insulin resistance suppressed in the PCA and EU group. However, these gut microbiota were negatively correlated with IgG or IgM. Additionally, genera, including *Rothia*, negatively correlated with IL-6 or insulin resistance showed a relative increase in the PCA and EU group, and these gut microbiota were positively correlated with IgG or IgM (Figure 5).

## 4. Discussion

An undesirable lighter weight at birth results in reduced viability of the offspring, leading to increased pre-weaning mortality rates [31]. It has been noted that the genetics-selected hyperprolific profiles of sows are a major contributing factor to the occurrence of low-viability piglets (Duroc × Landrace × Yorkshire) and neonatal mortality [32]. In comparison to the control group (CTL), supplementation with PCA and EU significantly enhanced litter weight at birth (21.93 ± 4.05 vs. 23.08 ± 4.31 vs. 17.94 ± 4.22; PCA vs. EU vs. CTL) and the number of healthy piglets (13.9 ± 3.75 vs. 14.1 ± 2.56 vs. 10.6 ± 2.59; PCA vs. EU vs. CTL) but with modest effects on average individual weight (1.39 ± 0.23 vs. 1.36 ± 0.32 vs. 1.31 ± 0.22; PCA vs. EU vs. CTL) or total litter size (16.2 ± 3.99 vs. 17.4 ± 3.72 vs. 13.9 ± 3.67; PCA vs. EU vs. CTL), suggesting that normal individual weights can be maintained even with larger litter sizes [33]. Appropriate interventions during late pregnancy can enhance the survival rates of low-birthweight piglets. Otherwise, an increase in total litter size may prove detrimental to their health outcomes. Notably, collaborative efforts aimed at improving both birthweights and colostrum quality play a crucial role in ensuring piglet survival from birth through weaning [34]. Supporting this notion, the addition of PCA significantly elevated IgG levels in colostrum, while supplementation with EU notably increased IgM concentrations in colostrum.

The results of improvements in birth weight, the number of healthy piglets, and colostrum immunoglobulin content following the addition of PCA and EU were consistent with the effects observed from supplementation with other bioactive substances, such as conjugated linoleic acids [35,36] and various polyphenols [37]. This suggests a potential influence of PCA and EU on the immunity-related status of sows that is transmitted to their offspring [15]. Given that chronic low-grade inflammation contributes to insulin resistance [38], our findings strongly support that the inclusion of PCA improved glucose metabolism and reduced inflammatory cytokine production in sows during late pregnancy. This aligns with our recent research indicating that PCA enhanced the expression of genes associated with insulin action, particularly *Fgf1*, *Igfbp2*, *Irs1*, and *Irs2* [26]. Unexpectedly, EU did not demonstrate significant improvement in glucose metabolism within this study. This might be attributed to its lower PCA content (0.018%), which was less than reported concentrations in other studies (0.04–0.05%) [39]. It is critical to note that the physiological effects observed in the PCA and EU groups likely stem from distinct mechanisms, given the vast difference in the delivered dose of protocatechuic acid. The effects of the high-dose, pure PCA are directly attributable to this compound. In contrast, the benefits of the EU extract, which delivered a negligible amount of PCA, are most likely mediated by other bioactive constituents (e.g., chlorogenic acid, a well-documented polyphenol known for its bioactive regulatory effects on inflammation [40,41,42].) present in the complex phytochemical matrix or through synergistic effects among them.

Higher levels of IgG coupled with lower levels of IgM or IgA constitute critical components of colostrum immune substances [43]. As an immunity-enhancing substance transmitted prenatally to offspring via either the placenta or yolk sac in sows [44], colostrum IgG levels can increase even at lower systemic inflammation levels [45] when supplemented with PCA. This finding was further supported by correlations observed between gut microbiota and immunoglobulins alongside serum indicators. Consequently, our previous study demonstrated that PCA mitigated inflammation in LPS-induced piglets (Pig Improvement Company line 337 × C48) [27]. Additionally, this study confirmed that both PCA and EU significantly decreased serum IL-6 levels. Another investigation also indicated that PCA inhibited IgG leakage in ischemia-induced rats [46]. The dietary EU utilized in this study demonstrated a significant enhancement in reproductive performance, alongside an increase in colostrum IgM.

Gut microbiota play crucial roles in maintaining metabolic balance during pregnancy in sows. This notion is supported by studies demonstrating that dietary interventions, such as liquid whey-enriched diets, can significantly reshape the gut microbial community and improve intestinal health in pigs [47,48]. The Firmicutes/Bacteroidetes ratio has been proposed as a potential biomarker for an imbalanced microbial community, which is often associated with inflammatory and metabolic disorders [49]. Evidence indicates that Firmicutes are more effective than Bacteroidetes as an energy source, promoting more efficient caloric absorption [27]. In the current study, although PCA rather than EU supplementation significantly increased the abundance of Bacteroidetes, the Firmicutes/Bacteroidetes ratio only exhibited a non-significant trend towards a decrease, which may be attributed to concurrent, non-significant variations within the Firmicutes phylum. This finding might elucidate their differing impacts on glucose metabolism. Furthermore, three genera were found to be significantly influenced by EU and PCA. For instance, *Caldicoprobacter*, identified as a primary candidate responsible for enhanced production of hydrolytic enzymes [50], was elevated by EU. *dgA11_gut_group*, a genus with limited research, showed increased levels due to PCA. Conversely, *Paraprevotella*, recognized as an opportunistic pathogen [51], was reduced by both EU and PCA.

Although the composition of colostrum is critical for piglets to obtain energy and maintain body temperature, sow-derived colostrum is characterized by a high content of protein but a relatively low content of lactose and fat [52]. In the present study, EU and PCA had limited effects on the number of somatic cells, fat, protein, lactose, degreased dry matter, total solids, and urea nitrogen in colostrum. However, several genera, including *Clostridium_sensu_stricto_3* and *Lysinibacillus*, decreased in the PCA and EU groups and were positively correlated with colostrum lactose and colostrum fat rather than the number of somatic cells, colostrum protein, degreased dry matter, total solids, and urea nitrogen, while genera, such as *Caldicoprobacter* and *Rikenellaceae_RC9_gut_group*, exhibited a reversed relevance. These results might offer an explanation that PCA and EU improved maternal immunostimulatory response in colostrum, potentially through modulating gut microbiota, thus improving the metabolic status of sows during late pregnancy. These genera belonged to different phyla and occupied a relatively considerable quantity, even though most of them had no statistical significance in the EU and PCA groups.

## 5. Conclusions

Dietary supplementation of PCA (200 g/t) and complex EU (1000 g/t) in sows during late pregnancy can differentially and effectively improve reproductive performance and colostrum immunoglobulin contents, at least partly by regulating inflammation and glucose metabolism-related microbial community structure.

## Figures and Tables

**Figure 1 animals-15-03166-f001:**
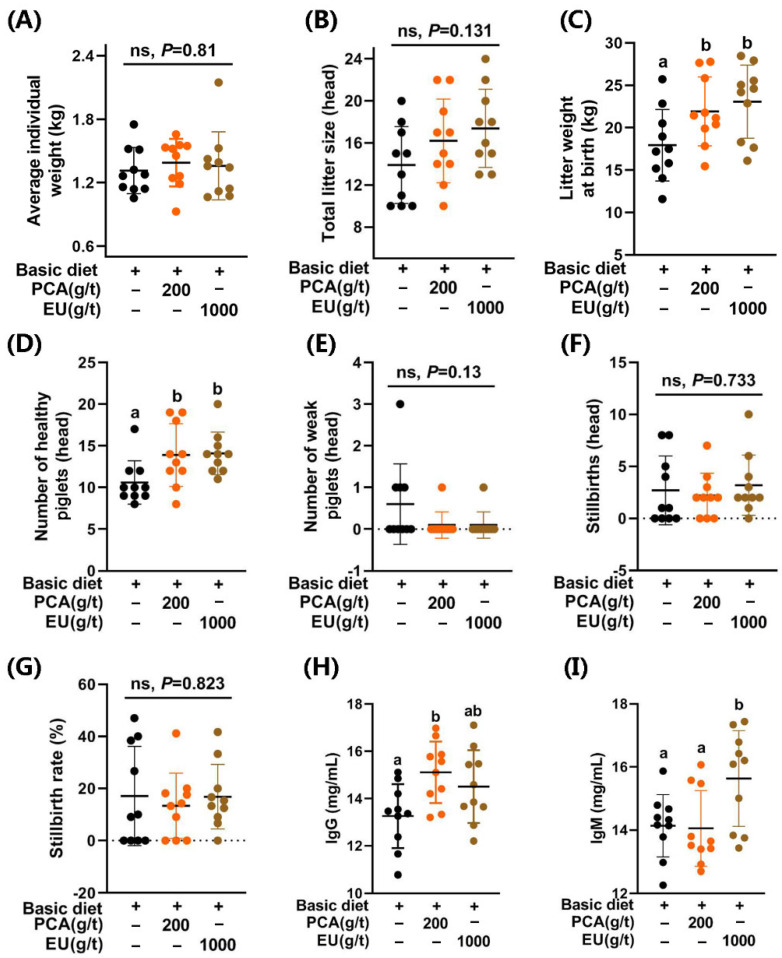
Effects of PCA and EU on reproductive performance and immunoglobulins in sows during late pregnancy. (**A**) Average individual weight. (**B**) Total litter size. (**C**) Litter weight at birth. (**D**) Number of healthy piglets. (**E**) Number of weak piglets. (**F**) Stillbirths. (**G**) Stillbirth rate. Contents of IgG (**H**) and IgM (**I**) in colostrum. Data are represented as mean ± SD (*n* = 10). Bars with different letters differ significantly (*p* < 0.05). EU, *Eucommia ulmoides* leaf extract; PCA, protocatechuic acid. Different lowercase letters above the bars indicate statistically significant differences among groups (*p* < 0.05), ns represents a non-significant difference (*p* ≥ 0.05).

**Figure 2 animals-15-03166-f002:**
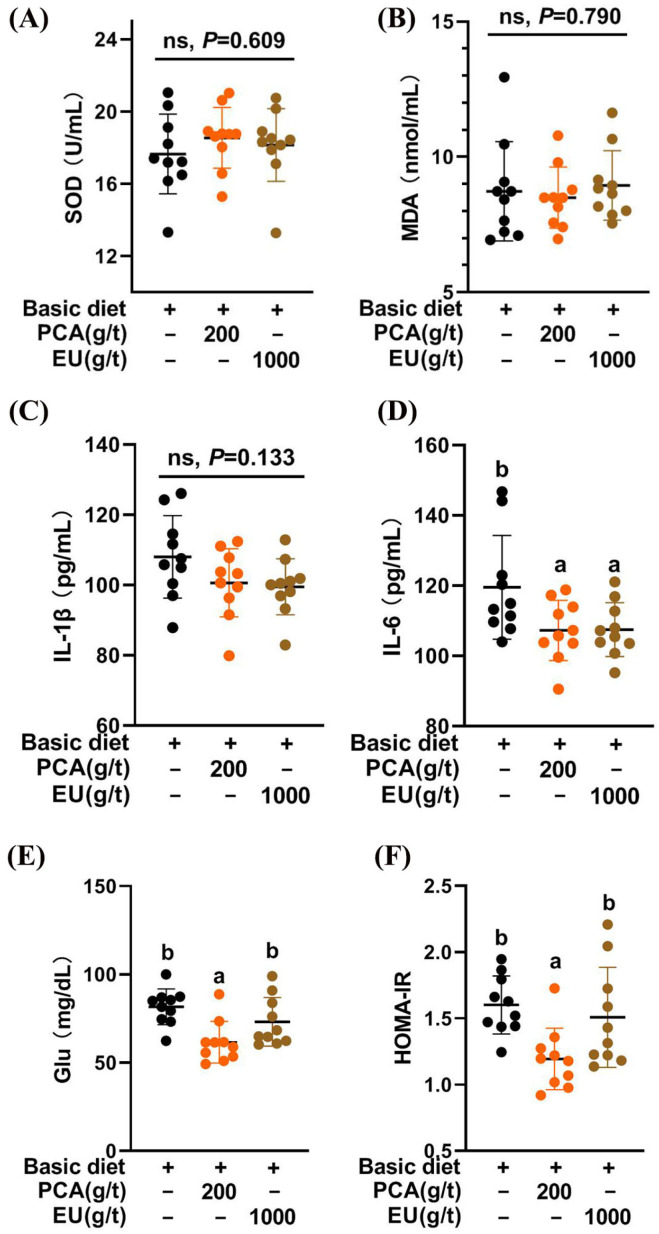
Effects of PCA and EU on redox status, inflammatory cytokines, and glucose metabolism in sows during late pregnancy. (**A**) Levels of SOD in serum. (**B**) Levels of MDA in serum. (**C**) Levels of IL-1β in serum. (**D**) Levels of IL-6 in serum. (**E**) Levels of glucose in serum. (**F**) Insulin resistance in serum. EU, *Eucommia ulmoides* leaf extract; Glu, glucose; HOMA-IR, homeostasis model assessment-estimated insulin resistance; MDA, malondialdehyde; PCA, protocatechuic acid; SOD, superoxide dismutase. Different lowercase letters above the bars indicate statistically significant differences among groups (*p* < 0.05), ns represents a non-significant difference (*p* ≥ 0.05).

**Figure 3 animals-15-03166-f003:**
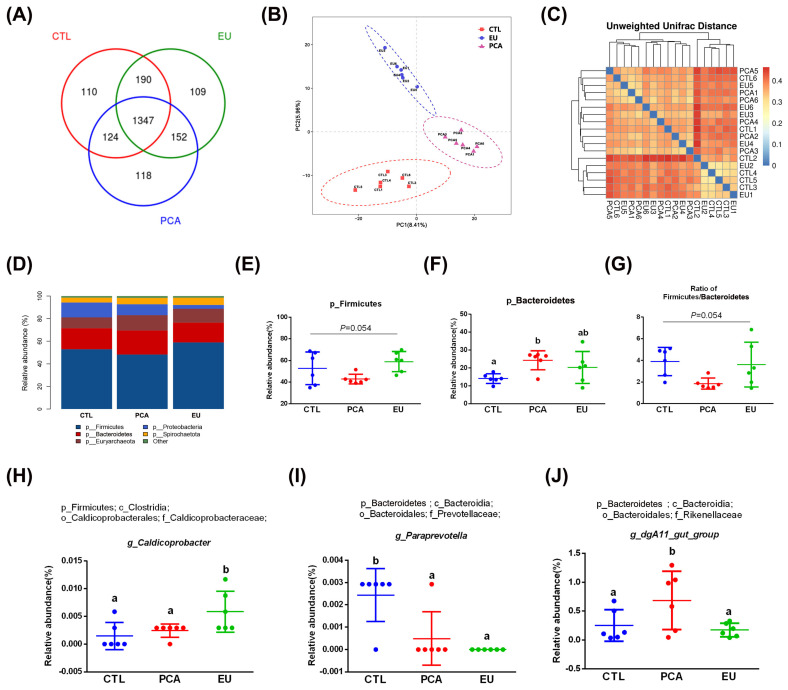
Effects of PCA and EU on gut microbiota in sows during late pregnancy. (**A**) Venn plot between the CTL, EU, and PCA groups. (**B**) PLS-DA. (**C**) β-diversity indices of unweighted_unifrac_distance. The redder the color, the farther the distance. (**D**) Relative abundance of gut microbiota at phylum levels. Relative abundance of Firmicutes (**E**) and Bacteroidetes (**F**). (**G**) Ratio of Firmicutes to Bacteroidetes. Relative abundance of *Caldicoprobacter* (**H**), *Paraprevotella* (**I**), and *dgA11_gut_group* (**J**). EU, *Eucommia ulmoides* leaf extract; PCA, protocatechuic acid; PLS-DA, partial least squares discrimination analysis. Different lowercase letters above the bars indicate statistically significant differences among groups (*p* < 0.05).

**Figure 4 animals-15-03166-f004:**
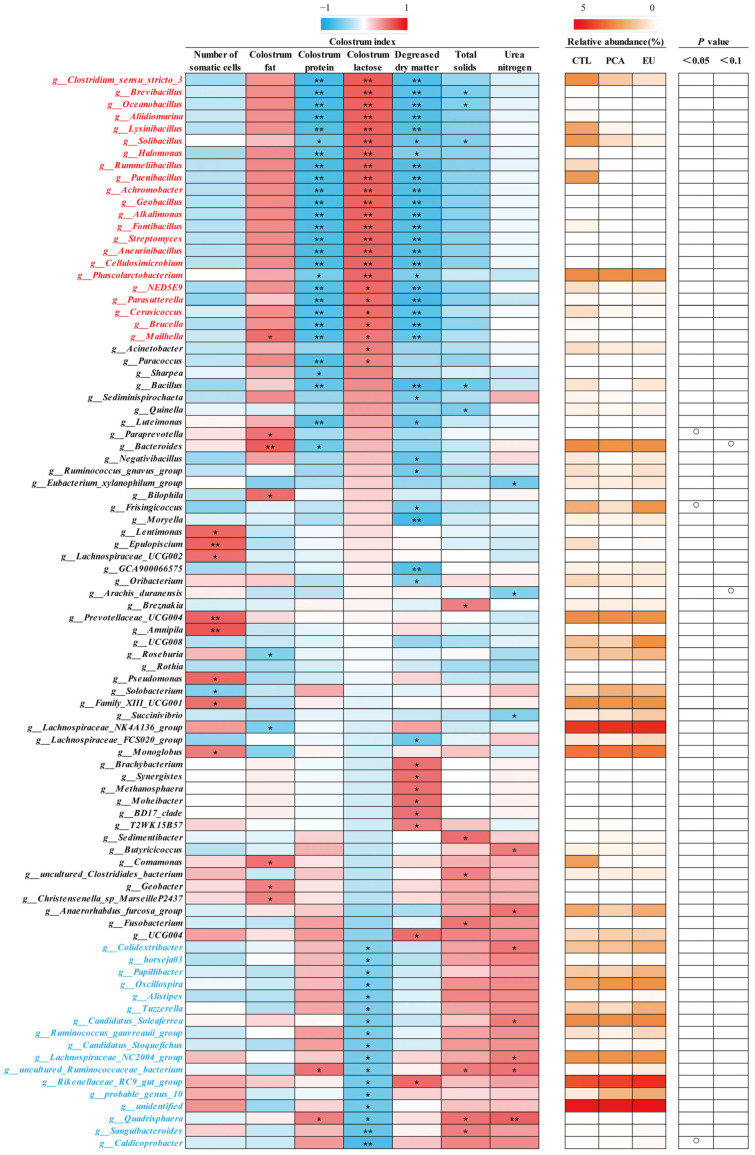
Correlation of genera and colostrum indicators. The intensity of the colors represents the degree of association (red, positive correlation; blue, negative correlation) or represents the relative abundance of a specific genus in CTL, PCA, and EU groups from 0–5% (from white to red). Significant correlations were found at * *p* < 0.05, ** *p* < 0.01, which represent in the form of a circle. EU, *Eucommia ulmoides* leaf extract; PCA, protocatechuic acid.

**Figure 5 animals-15-03166-f005:**
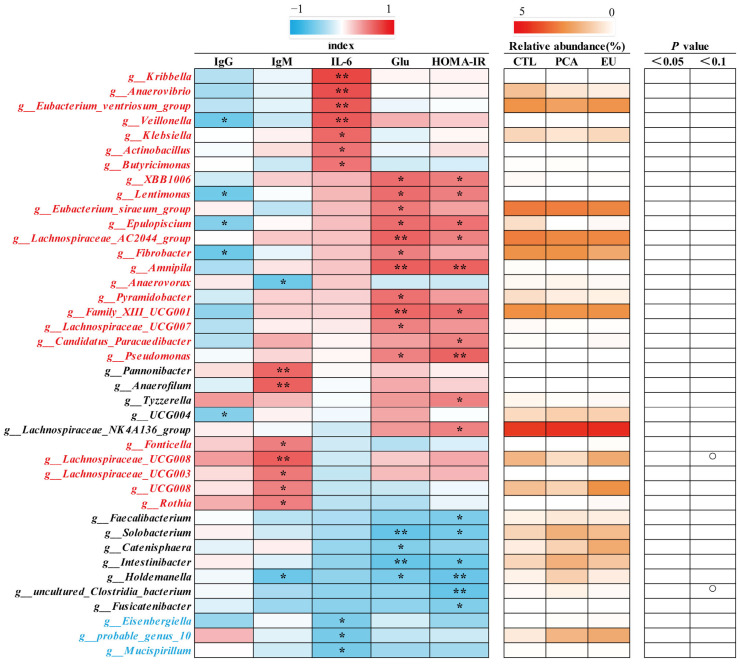
Correlation of genera and immunoglobulins together with serum indicators. The intensity of the colors represents the degree of association (red, positive correlation; blue, negative correlation) or represents the relative abundance of a specific genus in CTL, PCA, and EU groups from 0–5% (from white to red). Significant correlations were found at * *p* < 0.05, ** *p* < 0.01, which represent in the form of a circle. EU, *Eucommia ulmoides* leaf extract; Glu, glucose; HOMA-IR, homeostasis model assessment-estimated insulin resistance; PCA, protocatechuic acid.

## Data Availability

The 16S rRNA sequencing data generated in this study are openly available in the NCBI Sequence Read Archive (SRA) at https://www.ncbi.nlm.nih.gov/sra/, accessed on 15 October 2025, under the BioProject accession number PRJNA1345594. All other data supporting the findings are available within the article and its Appendix A.

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
