# Peer review of "Dietary Supplementation with Protocatechuic Acid and a Complex *Eucommia ulmoides* Leaf Extract Differentially Enhanced Reproductive Performance and Modulated Gut Microbiota in Late-Pregnancy Sows"

_animals, 2025, doi:10.3390/ani15213166_

Round 1

Reviewer 1 Report

Comments and Suggestions for Authors

Comments and suggestions for the authors are provided in the attached PDF.

Reviewer 2 Report

Comments and Suggestions for Authors

Abstract

L33: ' in late gestation' include the age and body weights of the sow for context

L34: 'n=10' there should be spaces in betweeen

Introduction

L76: Format 'Crusell et al.' correctly to the journal syle.

L77: 'decline in microbial diversity from late pregnancy to postpartum' provide the species and treatment of the trial being discussed so that readers can get a grip of the context.

L86-88: 'Anthocyanins, due to their catecholic structure, exhibit potent antioxidant properties and hold promise as therapeutic agents for metabolic disorders' go ahead and indicate the species and breed for better context

L90: 'and and' delete one

L97: 'piglets' include the breed

Materials and methods

L102: Reviewing the supplemental Table 1, only the control diet was detailed with the calculated values of the nutrients. Rather the supplemental Table 1, should present 3 diets (CTL, PCA, and EU) and further show analysed nutrient values especially to indicate the actual values of the PCA available in diet 2 and 3. Hence, the experimental diets which is a major factor in the trial needs to be analysed for clarity of the results.

L115-116: 'Spectrophotometric detection was done at 254 nm' indicate the spectrophotometer model, supplier, and location.

L121: 'sows with similar body conditions during late pregnancy' indicate their breed and average body weight.

L122: 'allocated into 3 treatments according to the days of pregnancy' were the days of pregnancy not uniform? you already indicate 80 days. Be consistent. if the days of pregnancy vary then give their range for clarity.

L123: 'barn' indicate the dimension of the barn. This will help the readers known the spacing allocated to the sows.

L125: 'pens' indicate the dimension

L126: 'special' Animals is an open access journal. Specify the model, supplier, and location.

L127: 'Then, fasting auricular venous blood and colostrum were collected once sows were delivering' do you mean during delivery? Because the sentence is not clear enough, so revise.

L119-128: Please specify the lighting program, humidity, temperature, of the house.

L151-156: It would be better to provide a table of the primer sequences used in this study for clarity and reproducibility.

Results
L185-186: specify the body condition score of each treatment.

L189: 'As shown in Figure 1A-G' 

Discussion

L310: 'piglets' state the breed

Reviewer 3 Report

Comments and Suggestions for Authors

I have several questions about the methodology.

All the links used in this article are relevant.

This article makes a good impression, but there are points that need to be clarified or supplemented:

  1. It is necessary to define the purpose of the study more clearly.
  2. It is necessary to explain the reason for the dosage of the introduced components in the diet.
  3. Provide the equipment used to determine the content of immunoglobulins and the method of determination
  4. Specify the equipment for determining the biochemical parameters of blood serum.
  5. The methodology does not specify when fecal samples were taken for microflora analysis.
  6. Which program was used for statistical calculations of experimental data.
  7. There are 2 identical paragraphs in lines 189-197.

I believe that this work can be published after correcting errors and editing.

Round 2

Reviewer 1 Report

Comments and Suggestions for Authors

The authors have addressed the major points raised in the previous review round. The suggested revisions have been incorporated into the manuscript and improve the clarity, transparency, and scientific interpretation of the study. I have only minor stylistic remarks (e.g., wording in a few places and the optional possibility of quantifying the PCA dose difference numerically), which do not affect the main conclusions. Overall, the manuscript is now substantially strengthened and is ready for publication.

Reviewer 2 Report

Comments and Suggestions for Authors

This is a well-articulated response note. I am satisfied.